# Hydration Fingerprints: A Reproducible Protocol for Accurate Water Uptake in Anion-Exchange Membranes

**DOI:** 10.3390/membranes15090257

**Published:** 2025-08-28

**Authors:** Sandra Elisabeth Temmel, Daniel Ölschläger, Ralf Wörner

**Affiliations:** Institute of Sustainable Energy Technology and Mobility (INEM), University of Applied Sciences Esslingen, 73037 Göppingen, Germany; daoegu00@hs-esslingen.de (D.Ö.); ralf.woerner@hs-esslingen.de (R.W.)

**Keywords:** water uptake, anion-exchange membranes, standardized blotting, hydration fingerprint, ATR-FTIR

## Abstract

Anion-exchange membranes (AEMs) not only enable the fabrication of catalyst-coated membranes without precious metals but are also projected to achieve a technology-readiness level (TRL) suitable for industrial deployment before the end of this decade. Accurate and reproducible water uptake data are essential for guiding AEM design, yet conventional gravimetric methods—relying on manual blotting and loosely defined drying steps—can introduce variabilities exceeding 20%. Here, we present a standardized protocol that transforms water uptake measurements from rough estimates into precise, comparable “hydration fingerprints.” By replacing manual wiping with a calibrated pressure-blotting rig (0.44 N cm^−2^ for 10 s twice) and verifying both dry and wet states via ATR-FTIR spectroscopy, we dramatically reduce scatter and align our FAAM-PK-75 (*Fumatech*, *Bietigheim*, *Germany*) results with published benchmarks in DI water, aqueous KOH (0.1–9 M), various alcohols, and controlled humidity (39–96% RH). These uptake profiles reveal how OH^−^ screening, thermal densification at 60 °C, and PEEK reinforcement govern equilibrium hydration. A low-cost salt-bath method for vapor-phase sorption further distinguishes reinforced from unreinforced architectures. Extending the workflow to additional commercial and custom membranes confirms its broad applicability. Ultimately, this work establishes a new benchmark for AEM hydration testing and provides a predictive toolkit for correlating water content with conductivity, dimensional stability, and membrane–ink interactions during catalyst-coated membrane fabrication.

## 1. Introduction

### 1.1. Purpose of the Study

Decarbonizing hard-to-electrify sectors demands not only renewable electricity but also robust methods to store, convert, and transport energy in chemical form. Water electrolysis delivers green hydrogen directly from water, positioning it as a key method for clean fuels and industrial feedstocks. Anion-exchange membrane (AEM) electrolyzers have emerged as a next-generation solution to meet the growing demand for sustainable hydrogen. Operating in alkaline media, they replace platinum-group catalysts with earth-abundant metals (e.g., nickel, iron), reducing material criticality and accelerating the oxygen evolution reaction. This lowers costs and improves cell performance [1,2,3]. Furthermore, AEMs are also poised to benefit from stricter PFAS regulations under EU REACH. However, the performance of these devices hinges, among other factors, on membrane hydration: water uptake and swelling directly govern hydroxide-ion conductivity, dimensional stability, and catalyst-layer morphology. Most current protocols use PEM standards (e.g., ANIONE [4]) that overlook hydroxide-specific solvation effects [5] and leave blotting and drying steps vaguely defined, introducing a variability of ±20% or more [6]. Without validated “dry” and “wet” reference states, comparing data across studies or labs remains unreliable. While advanced methods, such as dynamic vapor sorption (DVS), offer detailed insights into vapor-phase hydration behavior, they require specialized equipment and long equilibration times and are less accessible for routine membrane screening. To address this, we introduce the first fully standardized and transparent protocol for measuring water uptake in AEMs. Every step—drying, blotting, and soaking—is described in detail, and ATR-FTIR verification confirms both the hydration state and drying completeness. By systematically varying the medium, temperature, and relative humidity, practical “hydration fingerprints” are generated, laying the groundwork for consistent characterization, accelerated membrane development, and seamless translation from academia to industry.

### 1.2. Background and State-of-the-Art

Accurate, reproducible measurement of water uptake and swelling in AEMs is crucial for predicting hydroxide conductivity, dimensional stability, and long-term durability in alkaline electrochemical systems. Yet a survey of the literature reveals alarming discrepancies—even for the same commercial materials under similar conditions. Recent studies highlight this divergence. Khalid et al. [6] report a 35% thickness increase in FM-FAA3-50 at 60 °C in deionized water—derived from images and permeability testing rather than gravimetric methods—and note PK-75 damage after vacuum drying, underscoring conditioning artifacts. By contrast, Wijaya’s review [7] finds only 23–30 wt% uptake for FAA3-50, while Lee et al. [8] measure 59% uptake in FAA-3 at 70 °C but omit critical details regarding the soaking medium, drying, and blotting. Vandiver et al. [9] distinguish vapor-phase sorption (dynamic vapor sorption, DVS) from liquid-phase swelling in FAA-PEEK yet do not specify the sample preparation parameters. Commercial datasheets mirror these gaps: Fumatech’s FAAM-PK-75 claims 40–60% hydration (and 40% thickness gain) in 1M KOH at 20 °C [10] but provides no replication or handling guidance. Vapor-sorption studies similarly lack consistency. Zheng et al. [11] employed stepwise relative humidity (RH) environments to probe hydration kinetics in custom poly(phenylene oxide) (PPO)- and ethylene tetrafluoroethylene (ETFE)-based AEMs—revealing multi-stage uptake behavior and hysteresis—but did not perform liquid-phase tests or include commercial membranes. Zhang et al. [12] evaluated quaternary phosphonium–trimethylpiperidinium (QPTTP-x) copolymers by soaking them in deionized water at 30–80 °C, observing monotonic uptake increases with temperature; however, they also failed to disclose blotting pressure, drying time, or equilibrium criteria. Duan et al. [13] reported 10.0 wt% at 50 °C and 11.5 wt% at 80 °C under saturated vapor in an A201 membrane followed by a higher, linearly temperature-dependent uptake after 24 h of liquid immersion—an elegant demonstration of Schroeder’s paradox in AEMs. Modeling efforts have likewise advanced our molecular-level understanding. Density functional theory (DFT) and molecular dynamics (MD) simulations by Luque Di Salvo et al. [5] and Tomasino et al. [14] elucidate water clustering around charged sites, the influence of ion-exchange capacity on hydration shells, and hydrogen-bonding networks within the polymer matrix. However, these computational studies do not attempt experimental validation, omit evaluations of commercial membranes, and lack instructions for measurement protocols—limiting their practical impact on standardized testing. Together, these empirical and theoretical reports expose a critical gap: no harmonized, AEM-specific protocol exists for drying, blotting, equilibration, or verification of “dry” and “wet” states. Here, we introduce a fully transparent, gravimetric workflow—complete with controlled blotting pressure, defined equilibration criteria, and detailed handling procedures—to produce reproducible hydration fingerprints across chemical compositions, media, temperatures, and humidities.

## 2. Materials and Methods

### 2.1. Materials

The primary membrane used was FM-FAAM-PK-75 (*Fumatech*, *Bietigheim*, *Germany*), which is a 75 µm thick polysulfone membrane functionalized with quaternary ammonium groups and reinforced with PEEK [10]. For cross-validation, unsupported Fumasep membranes FAA-M-20 and FAA-3-50 (*Fumatech*, *Bietigheim*, *Germany*) and a self-synthesized AEM were included. All membranes were handled in a clean-room environment, with no ion-exchange pretreatment prior to measurement. Ultrapure deionized water (≥18.2 MΩ·cm), analytical-grade or higher KOH solutions (0.1–9 M, *Sigma-Aldrich*, *Taufkirchen*, *Germany*), isopropanol and ethanol (*Th.Geyer*, *Renningen*, *Germany*), glycerol (*Carl Roth*, *Darmstadt*, *Germany*), and salts for humidity control (MgCl_2_, NaCl, KNO_3_; *VWR Chemicals*, *Darmstadt*, *Germany*) were used as received.

### 2.2. Water Uptake Determination

Sample preparation and drying: Membranes were punched into 17.26 mm × 13.33 mm rectangles (*Vaessen Creative*, *Nuth*, *Netherlands*) and, without any pretreatment, dried over silica gel (*Th.Geyer*, *Renningen*, *Germany*) at ambient temperature (RH 0–5%, BC06, *TROTEC*, *Heinsberg*, *Gernany*) for 72 h. This low-temperature protocol avoids the microstructural changes reported at 50 °C [8]. Dryness was confirmed by ATR-FTIR (IR Affinity-1S, *Shimadzu*, *Kyoto*, *Japan*) for one batch, ensuring the absence of water-related bands, before immediate weighing on a calibrated microbalance (±0.1 mg, *KERN*, *Balingen*, *Germany*). Soaking protocol: Dried samples were immersed for 24 h in the chosen medium—DI water, kalium hydroxide (KOH) solutions (0.1–9 M, Sigma-Aldrich), or organic solvents (isopropanol, ethanol, glycerol)—at either room temperature (in Petri dishes to prevent deformation) or elevated temperature (in sealed vials on an *Ohaus HB2DG (Nänikon*, *Switzerland*) heating block to guarantee uniform temperature distribution (see Appendix A)). Blotting and gravimetric analysis: Two blotting methods were compared:Conventional touch-blotting using Kimtech™ wipes, as routinely described in the literature [14,15,16].○Standardized pressure-blotting (Figure 1): Samples were sandwiched between two pre-cut Kimtech™ layers, covered with a rigid pressing cap, and compressed twice for 10 s using a 1 kg weight (~0.44 N cm^−2^). By fixing both pressure and dwell time, this protocol eliminates operator variability and replicates the uniform compression experienced in full-cell assemblies—preventing local deformation while ensuring consistent surface-water removal. All materials were prepared in advance, and the membrane was weighed immediately after blotting using a balance positioned directly next to the setup, minimizing time-dependent variation due to ambient humidity or temperature. In addition, all handling was performed in a clean room with controlled environmental conditions to limit CO_2_ exposure, which can affect the ionic form of the membrane.

After blotting, wet mass (m_wet_) was recorded and water uptake was calculated asWU = (m_wet_ − m_dry_)/m_dry_(1)
where m_dry_ is the pre-soaking mass dried on silica gel assuming the weight gain was equal to the weight of the liquid absorbed by the membrane. The complete water uptake workflow, from sample cutting and drying to soaking, blotting, and gravimetric analysis, is illustrated in Figure 2.

### 2.3. Water Uptake Protocol Variations

To rigorously assess reproducibility and versatility, the water uptake protocol was subjected to a series of probing experiments in which the blotting method, electrolyte strength, temperature, swelling medium, and humidity were each varied. First, conventional manual blotting was directly compared to the calibrated pressure-blotting rig in DI water at 23 °C over 24 h. Next, water uptake was mapped across a KOH concentration gradient (0.1–9 M) under identical conditions, and swelling behavior was evaluated at both 23 °C and 60 °C to capture thermal effects. To emulate catalyst-coating environments, membranes were immersed in isopropanol, ethanol, and a 50:50 water–glycerol blend, while vapor-phase sorption over saturated salt solutions (33–93% RH) was employed to simulate gas-exposure scenarios. Cross-validation on three distinct membranes—FAA-3-20, FAA-3-50, and a self-synthesized AEM—under standard conditions (1 M KOH, 24 h, 23 °C) confirmed broad applicability. All measurements were carried out in duplicate. In parallel, ATR-FTIR spectroscopy (*IRAffinity-1S*, *Shimadzu*, *Kyoto*, *Japan*, 4000–600 cm^−1^, 4 cm^−1^
*resolution*, *transmission mode*) was applied to one test row of FAAM-PK-75 samples in both hydrated (soaked in DI water at RT for 24 h) and dried states (over silica gel for 72 h). Particular attention was paid to the water-associated O–H stretching region (around 3400 cm^−1^) to verify both uptake and the completeness of drying achieved by the standardized blotting procedure. Collectively, these studies establish a systematic, application-relevant hydration workflow designed to accelerate the deployment of high-performance AEMs in next-generation energy systems. All protocol variations are summarized in Table 1.

## 3. Results and Discussion

Water uptake underpins both ion transport and mechanical stability in AEMs and thus governs electrolyzer performance and durability [17]. To disentangle the effects of sample handling and test conditions, FAAM-PK-75 was characterized via a fully validated gravimetric protocol (ATR-FTIR confirmation of dry and wet states). Uptake was then measured as a function of blotting technique, soaking medium, temperature, and humidity. Several counter-intuitive trends emerged—most notably, a reduction in water uptake at elevated temperatures—and this underscores the importance of rigorous method design. The following sections analyze these results in turn, compare them to published data, and discuss their implications for practical AEM electrolyzer operation.

### 3.1. Influence of Blotting on Water Uptake

The type and intensity of blotting exert a dramatic influence on measured water uptake, as illustrated in Figure 3 for samples soaked in DI water. In the conventional approach, blotting pressure can vary widely between operators: test row 1 (orange) and test row 2 (gray)—both using Kimtech™ wipes—yield markedly different uptakes despite identical membranes and soaking conditions. In contrast, the defined pressure-blotting setup (blue) delivers consistent values across multiple users (two users, each with two test rows). These operator-dependent differences inflate the scatter and lead to large standard deviations. For DI water-soaked membranes, uptake values of ~3–4 wt% can vary between roughly 2 wt% and 6 wt%, making it impossible to distinguish subtle trends or define a reliable “true” value. These data suggest that our controlled 0.44 N cm^−2^ blot reduces operator-to-operator variability by more than 80%, compared to conventional wiping methods. Such variability, however, also plagues the literature: for instance, the swelling of FM-FAA3-50 in DI water at 60 °C has been reported as approximately 35% by Khalid et al. [6], whereas Tham & Kim [16] observed only ~10% thickness increase. Wijaya’s review [7] further reports 23–30 wt% water uptake for FAA3-50, and Lee et al. [8] measured uptake as high as 59% in FAA-3 at 70 °C. This fourfold range in reported values underscores the urgent need for a standardized protocol. ATR-FTIR analysis (see Methods, Appendix A) shows that after our silica gel drying step, the O–H stretching band at ~3300 cm^−1^ [18,19] vanishes completely, confirming that both free surface water and loosely bound absorbed water have been removed. In contrast, when we apply only the controlled pressure-blotting, the 3300 cm^−1^ band remains, demonstrating that blotting eliminates surface moisture while preserving the water held within the polymer matrix.

### 3.2. Evaluation of Effect of KOH Concentration on Water Uptake

Water uptake, swelling characteristics, and conductivity of AEMs are typically evaluated in pure water—largely because existing protocols have been adapted from fuel-cell research. However, AEM water electrolyzers operate under entirely different conditions, namely in the presence of supported electrolytes such as aqueous KOH. To better mimic real-world operation, we measured PK-75’s water uptake not only in deionized (DI) water but also across KOH concentrations from 0.1 M to 9 M, as shown in Figure 4. The FAAM-PK-75 shows a pronounced “dip-and-rise” hydration response when different samples are soaked independently across the KOH series. In DI water, the membranes absorb roughly 9 wt% water. In contrast, immersion in 0.1 M KOH lowers uptake to about 5 wt%, after which the water content climbs steadily with increasing KOH concentration—surpassing the DI water value above 6 M and reaching a maximum of approximately 11 wt% in 9 M KOH. It is suggested that this biphasic trend reflects two competing ionic effects. In DI water, the membrane’s interior—packed with fixed quaternary ammonium sites and their Br^−^ counter-ions—creates a huge osmotic pressure gradient against the pure, ion-free bath, so pure H_2_O floods in to relieve that imbalance and drives uptake to ≈9 wt% (see Figure 5b). At low to moderate KOH concentrations, the added salt sharply reduces that osmotic gradient and the invading OH^−^ screens the fixed charges, collapsing the hydrophilic nano-domains and producing a pronounced uptake minimum (see Figure 5c). Boström et al. [20] observed a similar monotonic decline in water uptake for PBI-based AEMs up to 2 M KOH, attributing it to this same balance of osmotic and Donnan forces. Beyond ≈ 6 M, however, the Donnan exclusion barrier [21] (i.e., the electrostatic exclusion of co-ions by the membrane’s fixed charges, which must be overcome before salt and water can freely partition) weakens: K^+^ co-ions begin to penetrate alongside OH^−^, and the overwhelming external osmotic pressure then forces water (and salt) back into the nano-channels, re-expanding the domains. Najibah et al. [21] reported that PBI based membranes show an initial drop in conductivity (in line with reduced swelling) in KOH solutions of 0.5 M and then recover at 1 M when Donnan exclusion fades and salt uptake increases (see Figure 5d). Similarly, Khalid et al. [6] measured FAAM-PK-75 (both OH- and Cl-exchanged) swelling of ≈7 wt% in DI water at room temperature, which fell to ≈5 wt% in 1 M KOH, before rebounding at higher concentrations. The quantitative agreement across these independent studies—with different chemical compositions, measurement protocols, and property end-points—confirms that our standardized pressure-blot method faithfully captures the underlying interplay of screening and osmotic driving forces that governs AEM hydration.

### 3.3. Temperature-Dependent Water Uptake Behavior

Figure 6 compares the 24 h water uptake of FAAM-PK-75 in both DI water and 1 M KOH at 23 °C versus 60 °C. In DI water, uptake falls dramatically—approximately 9% at 23 °C down to 4.5% at 60 °C—whereas in 1 M KOH, hydration rises slightly from about 6.4% at 23 °C to 7.5% at 60 °C. Most AEM studies in the literature report the opposite trend [17]. For example, Duan et al. [13] measured a polyamine-based AEM in pure water and saw a steady increase in uptake from 25 °C to 80 °C. Ahmed et al. [22] observed higher proton-exchange membrane hydration at elevated temperatures in acidic environments. Pandey et al. [19] and Vandiver et al. [9] used vapor-phase sorption (DI water) under controlled relative humidity and noted increased water content at higher temperatures (60 °C compared to 30 °C) under a given humidity. In DI water, PK-75 therefore contradicts these literature trends by losing nearly half its uptake at 60 °C. Thermal analysis might help explain this behavior: TGA (Appendix A) shows <1% mass loss up to 200 °C, and DSC (Appendix A) reveals only a weak, broad transition below ≈150 °C, making chemical degradation unlikely. Instead, it is suggested that warming to 60 °C provides just enough segmental mobility for the polysulfone backbone to relax into a slightly denser packing [14]. This “thermal tightening” would shrink the hydrophilic nano-domains, reducing the free volume available—so even though diffusion is faster at 60 °C, there simply is not enough space for as many water molecules, and uptake falls sharply in DI water (see Figure 5e). A directly analogous effect was observed in Nafion: when mildly annealed, its water content (λ) drops despite the temperature rise, as described by Kusoglu and Weber [23]. In 1 M KOH, however, the abundant external OH^−^ ions might already partially screen the quaternary ammonium sites at 23 °C, so the channels cannot swell as much to begin with (see Figure 5c, Section 3.3). Heating to 60 °C then could provide faster diffusion, which might just barely outweigh any further densification, resulting in a small net increase in uptake rather than a drop (see Figure 5f). At higher KOH concentrations (≥6 M, Section 3.2), screening and osmotic pressure combine to re-expand the channels, so uptake increases again—producing a minimum near 1 M. These findings, summarized schematically in Figure 6, illustrate why FAAM-PK-75 possibly behaves differently to many other AEMs in DI. It is also important to note that water uptake must be measured at the actual operating temperature and in the medium of interest, since both factors can drastically alter channel morphology. Potential follow-on SAXS/WAXS or DMA studies could confirm whether domain shrinking or modulus changes accompany the proposed thermal densification. These findings emphasize that temperature’s effects on FAAM-PK-75 hydration are highly dependent on the ionic environment and that operating above room temperature—even in 1 M KOH—can improve hydration (albeit less dramatically than in DI water). Consequently, FAAM-PK-75 may require extra humidification or a higher KOH concentration to maintain peak conductivity in an elevated-temperature electrolyzer.

### 3.4. Effect of Swelling Media on Water Uptake Behavior

To simulate conditions encountered during catalyst-coated membrane (CCM) fabrication—where uncontrolled expansion can compromise catalyst adhesion—FAAM-PK-75 samples were soaked for 24 h at 23 °C in deionized (DI) water, isopropanol (IPA; 2-propanol), ethanol (EtOH), and a 50:50 wt% glycerol/DI water blend, then pressure-blotted using our standardized protocol. As shown in Figure 7, DI water produces the highest uptake (~9 wt%), whereas all organic solvents induce a much lower uptake: IPA (~1.5 wt%), glycerol/DI (~4 wt%), and EtOH (~5 wt%). These rankings (IPA < EtOH < DI water) echo findings in the literature for ion-exchange membranes in alcohols. Laín and Barragán [15] measured four commercial AEMs at 30 °C in water, methanol (MeOH), EtOH, and 1-propanol (IPA). They explained that heterogeneous AEMs swell more in aqueous mixtures than in neat alcohols because their larger, more tortuous hydrophilic domains absorb water readily, while longer-chain or more viscous alcohols form larger hydrogen-bonded aggregates that cannot penetrate those domains as easily—yielding the observed order (IPA ≈ 3–4 wt% < EtOH ≈ 5–6 wt% < H_2_O). Godino et al. [24] reported—(although for a proton-exchange membrane (PEM)) using Nafion 117 at 25 °C, tested both in pure alcohols and in the same alcohols containing 1 M KOH—that uptake follows the order IPA (~3.0 wt%) < 2-PrOH (~3.5 wt%) < MeOH (~5.1 wt%) < EtOH (~7.2 wt%), even at 1 M KOH. In this case, OH^−^ partial screening modestly collapses channels across all solvents but does not change the ranking. Yi and Bae [25] found for PEMs at 273 K that swelling decreases as alcohol chain length and steric bulk increase (MeOH ≈ 132% > EtOH ≈ 90% > 1PA ≈ 60% > 2-PrOH ≈ 55%), attributing this to the lower polarity (dielectric constant ε) and increased steric hindrance reducing hydrogen-bonding affinity with sulfonic groups. Mechanistically, three factors dominate:Polarity (dielectric constant ε): DI water (ε ≈ 80) strongly solvates quaternary ammonium sites and OH^−^ counter-ions, generating high osmotic pressure and maximal swelling. EtOH (ε ≈ 24) has lower polarity, so it cannot solvate ionic sites as effectively—yielding moderate uptake (~5 wt%). IPA (ε ≈ 18) is less polar still, so it forms fewer hydrogen bonds with ionic centers, resulting in minimal uptake (~3 wt%). A lower dielectric constant means the solvent cannot screen fixed charges or stabilize ion clusters as well, so its affinity for sulfonic acid (or quaternary ammonium) sites is reduced and swelling is suppressed [26].Molecular size and viscosity: IPA’s branched structure and higher viscosity (≈2 cP at 20 °C) hinder its diffusion into sub-nanometer hydrophilic channels, limiting uptake. EtOH (≈1.2 cP) diffuses more readily, while glycerol/DI (ε ≈ 50–55; high viscosity from glycerol’s three –OH groups) swells moderately (~6 wt%) by partially hydrating without full domain expansion [15].Hydrogen-bonding affinity: Glycerol’s multiple –OH groups support strong hydrogen bond networks that partially hydrate and plasticize the polymer backbone. EtOH forms fewer hydrogen bonds, and IPA’s single –OH bond (in a branched environment) yields a weaker, less extensive network, reducing its ability to open ionic channels [15]. All relevant parameters of the solvents are summarized in Appendix A. From a practical standpoint, these uptake measurements could guide the selection of ink composition (e.g., ratio of DI to solvent—“dry ink” versus “liquid medium” ratios) or solvent type to minimize swelling and optimize the catalyst–membrane interface. In particular, pretreatment in any solvent immediately before ink application may help preserve membrane planarity, reduce delamination risk, and improve catalyst adhesion. Further studies—examining water uptake at different “DI to solvent ratios”, in different mixed-solvent systems, or for extended exposure times—are necessary to identify a more clear correlation between water uptake and the ideal ink formulation.

### 3.5. Effect of Relative Humidity on Water Uptake Behavior

Figure 8 shows FAAM-PK-75’s water uptake under three controlled-RH atmospheres (≈39%, 74% and 96% at 23 °C). Strikingly, the membrane absorbs a nearly constant ~5 wt% of water at all humidity levels. This near invariance suggests that the PEEK reinforcement mechanically caps the swelling of the hydrophilic network: regardless of how much water the vapor phase supplies, the stiff support prevents further domain expansion, locking in ~5 wt% hydration. These data suggest that PEEK reinforcement effectively limits vapor-phase swelling, although testing across a broader set of temperatures and chemical compositions will be needed to confirm the universality of this behavior. A prior study supports this interpretation—Vandiver et al. [9] likewise observed a ~5 wt% plateau in gas-phase uptake for a PEEK-reinforced AEM at 30 °C—whereas an unsupported, dual-domain ionomer showed monotonic uptake increases as RH rose [19]. To test the protocol’s sensitivity to structural differences, proof-of-concept studies were carried out on a PEEK-reinforced PEM (FM-F-990-PK) and on unreinforced Nafion™ (*The Chemours Company*, *Wilmington*, *NC, USA*) N117 (Appendix A). The reinforced PEM plateaued at ~2 wt%, while Nafion™ swelled from ~5 wt% at 39% RH to ~14 wt% at 96% RH—mirroring Pandey’s findings. These side experiments confirm that our simple, salt-controlled vapor-sorption protocol can discriminate between reinforced and unreinforced membranes. While these proof-of-concept studies demonstrate the protocol’s sensitivity to structural differences, additional experiments on diverse AEM formulations will be required to fully establish its discriminative power. Overall, this straightforward, low-cost screening tool offers a rapid estimate of vapor-phase hydration under both water electrolysis conditions and coating or handling atmospheres. Extending these studies across a wider range of AEM compositions will be essential to fully validate their general applicability.

### 3.6. Cross-Validation with Other AEMs

To demonstrate the broad applicability of our method, we soaked four different membranes—FAAM-20, FAAM-PK-75, FAA-3-50 and a self-synthesized AEM—in 1 M KOH for 24 h at 23 °C, then applied the standardized pressure-blotting procedure. As shown in Figure 9, water uptake increases monotonically from FAAM-20 (≈12 ± 1 wt%) to FAAM-PK-75 (≈24 ± 2 wt%), FAA-3-50 (≈57 ± 3 wt%), and finally the self-synthesized AEM (≈62 ± 4 wt%). Importantly, the ordering of PK-75 below FAA-3-50 mirrors independent findings: Khan et al. [26] reported ≈8 wt% uptake for PK-75 versus ≈22 wt% for FAA-3-50 in 1 M KOH, and Najibah et al. [21] observed PK-75 uptakes of ≈15 wt% (0.1 M KOH) and ≈25 wt% (1 M KOH) compared to FAA-3-50’s ≈ 50 wt% and ≈55 wt% under the same conditions. Even though Fumatech’s datasheets [10,27,28] report greater absolute swelling in 9–12 M KOH, they preserve the same hydration hierarchy (FAAM-20 < PK-75 < FAA-3-50—see Appendix A). These cross-validation tests confirm that our controlled pressure-blotting protocol not only reproduces absolute uptake values with low scatter (SD < 5%) but also faithfully captures the relative water absorption trends across chemically diverse AEMs.

## 4. Conclusions

We have developed and validated a robust, reproducible gravimetric protocol for measuring water uptake in AEMs. By replacing ambiguous, operator-dependent blotting with a defined 0.44 N cm^−2^ pressure blot and confirming complete drying via ATR-FTIR, we have eliminated the single largest source of scatter in conventional methods while also reproducing key uptake benchmarks for FAAM-20, FAA-3-50, and FAAM-PK-75. The baseline sequence is straightforward: first, membranes are dried over silica gel for 72 h; next, they are immersed in 1 M KOH at 23 °C for 24 h; finally, a 0.44 N cm^−2^ pressure blot is applied twice for 30 s to define the “wet” mass. Under these controlled conditions, each water uptake profile becomes a unique hydration fingerprint, revealing how four factors govern equilibrium hydration:KOH concentration: Uptake drops from ≈9 wt% in DI water to ≈6 wt% at 1 M KOH (due to OH^−^ screening), then increases to ≈11 wt% at 9 M as osmotic swelling overtakes screening.Swelling medium: Organic solvents are ranked as isopropanol (~1.5 wt%) < glycerol/DI (~4 wt%) < ethanol (~5 wt%) < DI water (~9 wt%), showing IPA as a promising low-swelling pre-soaking medium for ink compatibility.Temperature: Heating from 23 °C to 60 °C roughly halves uptake in DI water (thermal densification), while in 1 M KOH, a modest +2% increase indicates that external OH^−^ partially counteracts densification.Relative humidity: Vapor-phase sorption over 39–96% RH yields a steady ≈5 wt% uptake for FAAM-PK-75, demonstrating that PEEK reinforcement inhibits gas-phase swelling and confirming that saturated salt environments offer a simple, reliable method for controlled-RH measurements.

These hydration fingerprints correlate directly with critical performance metrics—ionic conductivity, dimensional stability, and coating compatibility—positioning gravimetric water uptake measurements as a powerful tool for elucidating structure–property relationships in AEMs. While the baseline protocol readily identifies significant differences among membranes, variations in soaking medium, temperature, or equilibration time can be applied to investigate specific chemical compositions or operational scenarios in greater depth.

## 5. Outlook

Our standardized water uptake protocol paves the way for a range of follow-on studies. In the near future, the gravimetric + ATR-FTIR workflow could be applied directly to leading commercial AEMs (e.g., Sustainion^®^, Piperion^®^), as well as newly synthesized chemical compounds, allowing us to map how backbone structure, ion-exchange capacity and reinforcement strategies influence hydration across a range of materials. Introducing mixed-salt baths would allow us to fill the gap between 39 and 96% RH and generate a high-resolution gas-phase sorption isotherm. By coupling precise uptake curves with ionic conductivity measurements, it will be possible to uncover quantitative hydration–transport relationships, directly tying water content to hydroxide mobility. At the same time, systematic pre-conditioning experiments in various solvent blends—from simple alcohols to full catalyst ink formulations—could identify pre-soaking materials that virtually eliminate swelling during catalyst-coated membrane fabrication. Taken together, these straightforward, low-cost water uptake assays show the potential for a predictive toolbox for membrane design, process optimization, and the accelerated rollout of next-generation AEM electrolyzers.

## Figures and Tables

**Figure 1 membranes-15-00257-f001:**
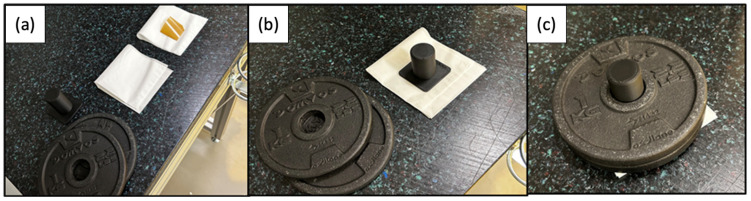
Stepwise setup of the standardized pressure-blotting procedure used for water uptake measurements in a clean room environment. (**a**) Blotting materials: membrane, Kimtech layers, pressing cap, and weights. (**b**) Setup with pressing cap and aligned Kimtech wipes. (**c**) Defined pressure applied using stacked weights equivalent to 0.44 N cm^−2^.

**Figure 2 membranes-15-00257-f002:**
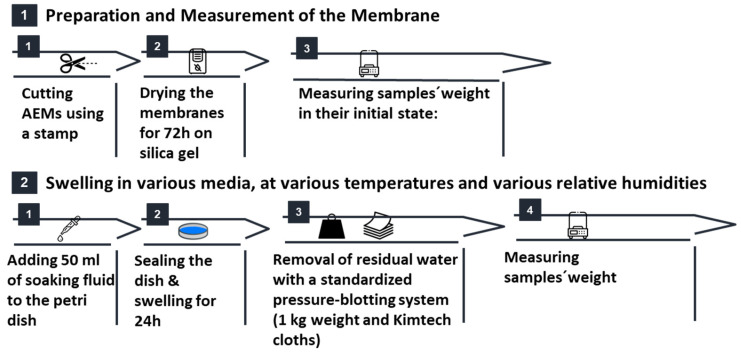
Water uptake workflow used in this study. Membranes are first cut to size, dried over silica gel, and weighed to record the dry mass. They are then soaked in the chosen medium (liquid or vapor) at the specified temperature or relative humidity. After soaking, surface water is removed by the standardized pressure-blotting protocol, and the wet mass is recorded.

**Figure 3 membranes-15-00257-f003:**
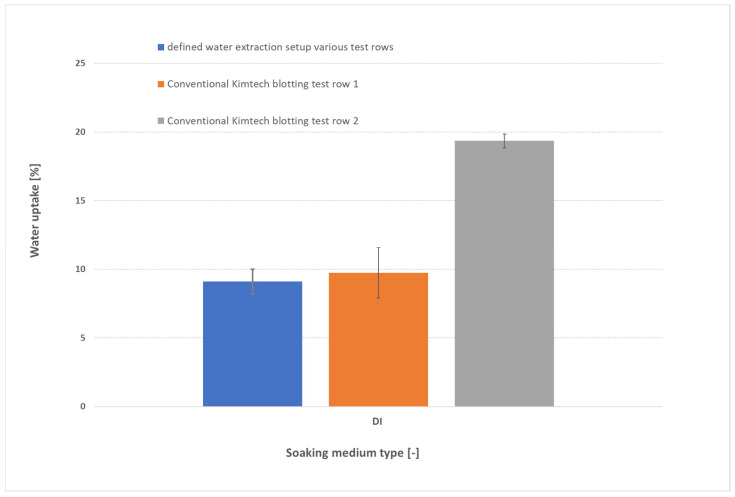
Comparison of water uptake measurements for FAAM-PK-75 membranes soaked in DI water (24 h, 23 °C, *n* = 2) using three blotting approaches: defined pressure-blotting setup by multiple users (blue) and conventional Kimtech™ touch-blotting by Person 1 (orange) and by Person 2 (gray). Error bars represent ±1 standard deviation. The data highlight the large operator-dependent variability of manual blotting versus the consistency afforded by the standardized pressure-blotting protocol.

**Figure 4 membranes-15-00257-f004:**
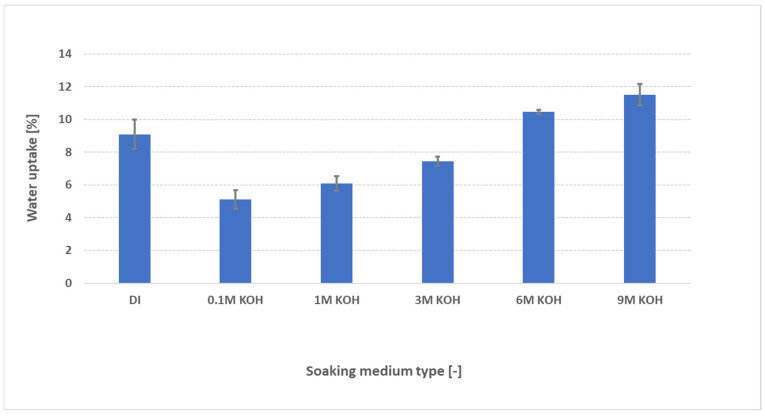
Water uptake of FAAM-PK-75 after 24 h of soaking at 23 °C in DI water and various KOH concentrations (0.1–9 M), measured using the standardized pressure-blotting protocol (*n* = 3). Bars indicate mean ± standard deviation. Uptake decreases initially from DI to 0.1 M KOH and then rises progressively at higher KOH molarities, reflecting the combined effects of OH^−^ screening and osmotic swelling.

**Figure 5 membranes-15-00257-f005:**
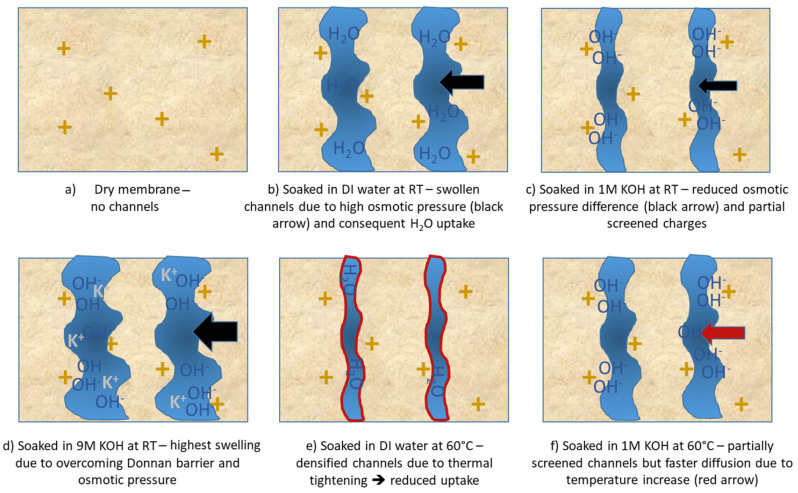
Schematic of FAAM-PK-75 hydrophilic domain morphology under different hydration conditions. In all panels, yellow “+” symbols denote fixed quaternary ammonium sites in the polymer; blue regions are water-filled nanochannels; and black “OH^−^” and green “K^+^” labels indicate internal and external ions, respectively. (**a**) Dry membrane—no channels: only quaternary sites are visible on a light-tan background. (**b**) DI water at 23 °C—swollen channels: wide blue domains illustrate maximal osmotic swelling by H_2_O diffusion (black arrows indicate driving osmotic pressure). (**c**) 1 M KOH at 23 °C—screened channels: external K^+^ and OH^−^ flood the domains, screening the “+” sites and collapsing channels (black arrows). (**d**) 9 M KOH at 23°C—higher osmotic pressure leads to the Donnan barrier being overcome and highest swelling. (**e**) DI water at 60 °C—densified channels: thermal tightening leads to lower water uptake. (**f**) 1 M KOH at 60 °C—partial screening with thermal diffusion: channels expand slightly versus panel (**d**) (red arrow indicates enhanced diffusion overcoming some screening), while the remaining ions are screened by abundant OH^−^.

**Figure 6 membranes-15-00257-f006:**
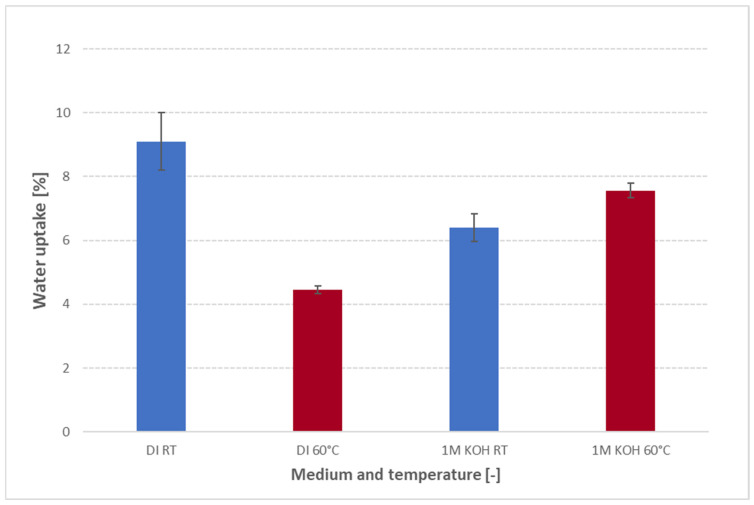
Water uptake of FAAM-PK-75 after 24 h of soaking in DI water and 1 M KOH at 23 °C (RT) and 60 °C, measured using the standardized pressure-blotting protocol (*n* = 2). Bars show mean ± standard deviation. In DI water, uptake drops sharply at 60 °C compared to RT, indicating thermal densification of hydrophilic domains. In 1 M KOH, uptake is higher at both temperatures due to screening by OH^−^, and it increases modestly at 60 °C as faster diffusion partially offsets densification.

**Figure 7 membranes-15-00257-f007:**
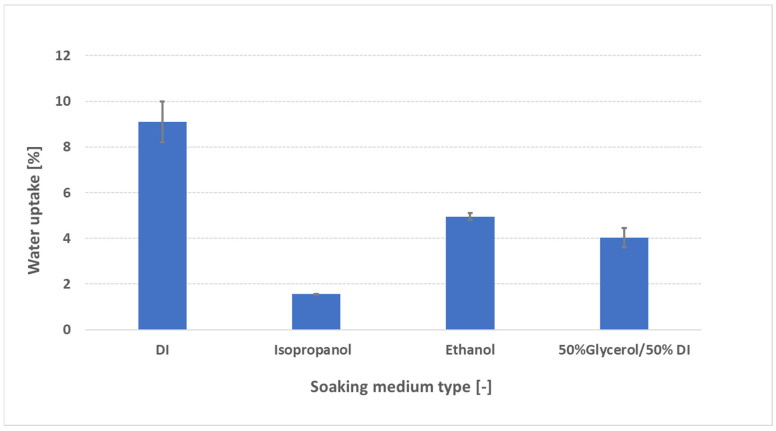
Water uptake of FAAM-PK-75 after 24 h of soaking in various media at 23 °C. Bars represent mean ± standard deviation (*n* = 2). The minimal uptake in isopropanol (IPA) suggests that using IPA as an ink solvent can reduce swelling, whereas DI water dramatically increases uptake.

**Figure 8 membranes-15-00257-f008:**
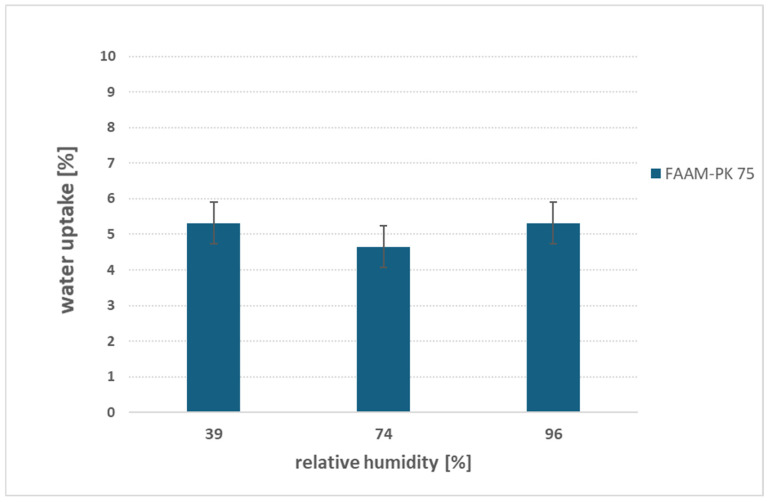
Gravimetric water uptake of FAAM-PK-75 exposed to controlled vapor-phase environments at 23 °C. Samples were suspended over saturated salt solutions to achieve ≈39%, 74%, and 96% RH for 48 h, then blotted under standardized pressure. Despite the increasing humidity, PK-75 consistently absorbs ~5 wt% water, indicating that the PEEK reinforcement mechanically limits the swelling of the hydrophilic domains.

**Figure 9 membranes-15-00257-f009:**
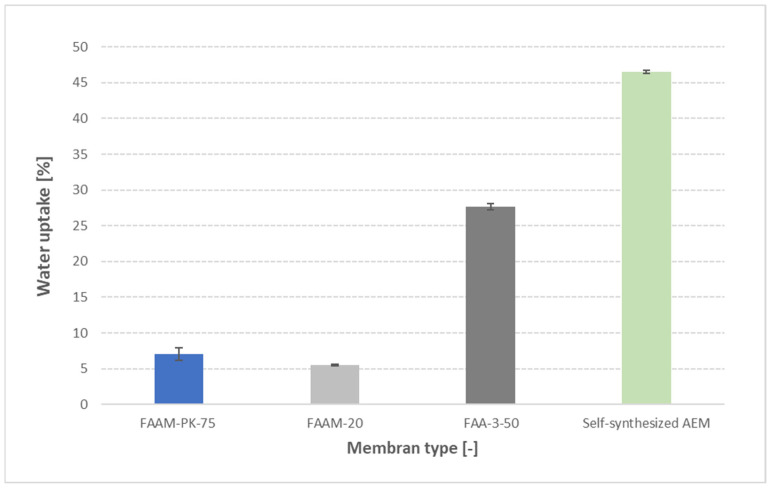
Water uptake of four AEMs at 23 °C after 24 h of soaking in DI water, followed by standardized pressure-blotting (mean ± standard deviation, *n* = 2). PK-75 exhibited 7 ± 0.5 wt%, FAA-M-20 exhibited 6 ± 0.4 wt%, FAA-3-50 exhibited 27 ± 1 wt%, and the self-synthesized AEM exhibited 46 ± 1 wt%.

**Table 1 membranes-15-00257-t001:** Summary of all water uptake protocol variations. Protocol name, membrane type, soaking medium, duration, temperature, and relative humidity (rH) are shown. For samples soaked in liquid, an rH of 100% was assumed.

Protocol	Membrane Type	Soaking Medium	Duration (h)	T (°C)	rH (%)
Blotting comparison	FAAM-PK-75	DI water	24	RT	100
KOH concentration effect	FAAM-PK-75	KOH solutions (0.1 M–9 M)	24	RT	100
Temperature-dependent swelling	FAAM-PK-75	DI water 1 M KOH	24	RT and 60	100
Alternative immersion media	FAAM-PK-75	Isopropanol Ethanol 50:50 H_2_O–glycerol	24	RT	100
RH-controlled sorption	FAAM-PK-75	Vapor over salts: MgCl_2_ (≈33% RH) NaCl (≈75% RH) KNO_3_ (≈93% RH)	48 h	RT	33/75/93
Cross-validation	-FAAM-20, -FAA-3-50, -self-synthesized AEM	1 M KOH	24	RT	100

## Data Availability

The original data presented in the study are openly available in FigShare at https://10.6084/m9.figshare.29571860.

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
