# Peer review of "Hydration Fingerprints: A Reproducible Protocol for Accurate Water Uptake in Anion-Exchange Membranes"

_membranes, 2025, doi:10.3390/membranes15090257_

Round 1

Reviewer 1 Report

Comments and Suggestions for Authors

This study is interesting. Normally, in the measurement for water uptake of ion exchange membranes, a large measurement error is caused by the test method and operational variability. The present work provides a stable solution for water uptake testing. However, compared to previous measurement methods, the present one is more complex and highly operationally demanding.

  1. Pressurization ~ 0.44 N/cm², did different pressures or pressurization methods affect the measurement results?
  2. Operating factor such as pressurization maintained for 10s, did the difference in maintenance time affect the water content results?
  3. How to ensure that the methodology in this manuscript is applicable in testing water uptake for other AEMs?
  4. One item of interest, could this method of water uptake testing be applied to the testing of ion exchange capacity?
  5. It is recommended that the figures be placed after the corresponding paragraph.

Author Response

1.Pressurization ~ 0.44 N/cm², did different pressures or pressurization methods affect the measurement results?

Thank you for your thoughtful question. While we did not conduct a systematic investigation into the effect of different pressures or pressurization methods on water uptake, we did perform preliminary trials to optimize the setup. During this phase, we tested various configurations and selected one that ensured ease of handling, uniform pressure distribution, and reproducible results.

2.Operating factor such as pressurization maintained for 10s, did the difference in maintenance time affect the water content results?

Thank you for this important observation. Our primary goal was to establish a simple and reproducible protocol for measuring water uptake. In our preliminary tests, we experimented with shorter pressurization times, including a single 10-second application. However, this proved insufficient, as residual water spots remained visible on the membrane surface, indicating incomplete absorption.

Once we identified a procedure that consistently resulted in a spot-free surface and reproducible measurements, we did not further investigate the effects of longer pressurization durations or multiple shorter intervals. This decision was made to keep the protocol as straightforward and practical as possible.

3. How to ensure that the methodology in this manuscript is applicable in testing water uptake for other AEMs?

Thank you for this relevant point. As demonstrated by our cross-validation results, the proposed protocol is applicable to other AEMs. In these tests, we applied the same methodology to different AEM samples and achieved consistent and reproducible outcomes. This indicates that the protocol is robust and not limited to a specific membrane type.

Of course, when applying the method to new materials, users should verify that the key assumptions—such as achieving a spot-free surface under the chosen pressurization conditions—are still met. Minor adjustments may be necessary depending on the specific properties of the AEM, but the general procedure provides a solid and adaptable framework.

4.One item of interest, could this method of water uptake testing be applied to the testing of ion exchange capacity?

This is an interesting perspective and a promising idea for future research. While our current study focuses on water uptake, it could be worthwhile to explore whether the same setup can be adapted for measuring ion exchange capacity (IEC). One potential approach would be to first determine the IEC using a conventional method, and then assess whether our protocol could be modified accordingly to yield comparable or complementary results.

Such an investigation would require careful adaptation and validation, and would likely form the basis of a separate, in-depth study. Nonetheless, we appreciate the suggestion and consider it a valuable direction for future work

5.It is recommended that the figures be placed after the corresponding paragraph

Thank you for the suggestion. We fully agree that placing figures directly after the corresponding paragraphs can enhance readability and comprehension. However, we followed the journal’s submission guidelines, which require figures to be placed in a separate, designated section at the end of the manuscript. To support clarity, we have ensured that all figures are clearly referenced and described within the main text.

Reviewer 2 Report

Comments and Suggestions for Authors

The manuscript presents a valuable standardized gravimetric protocol for measuring water uptake in anion-exchange membranes (AEMs), validated by ATR-FTIR and tested across diverse conditions. It addresses a critical gap in AEM characterization with promising reproducibility. However, several points need to be addressed to strengthen the manuscript before acceptance. Detailed comments are as follows:

  1. Given that AEMs contain reactive functional groups such as quaternary ammonium, has the potential interaction with ambient CO2 been considered? CO2 absorption could lead to the formation of bicarbonate or carbonate species, potentially altering the membrane's ionic form and affecting water uptake. How did the authors account for or minimize this effect during handling and measurement??
  2. When measuring water uptake under different temperature or humidity conditions, the membrane must be tested promptly after being removed from the dish to minimize the influence of ambient water vapor and room temperature on its mass. However, the standardized pressure-blotting method likely takes more time than conventional touch-blotting using Kimtech™ wipes. Has the potential error introduced during this additional handling time been considered by the authors? Could this error be comparable to or even larger than the differences between the two blotting methods themselves? What strategies could be employed to minimize this time-dependent error?
  3. The standardized pressure-blotting method uses ~0.44 N cm⁻² applied for 10 s. Has the sensitivity of water uptake to blotting pressure been evaluated? Specifically, how robust is the measured uptake to slight variations in applied pressure?
  4. In the introduction section, the authors are encouraged to include a brief discussion of more advanced characterization techniques for water uptake, such as dynamic gravimetric vapor sorption (DVS), which has been used to investigate water sorption behaviour and diffusion coefficients in AEMs. It would be beneficial to reference and cite relevant literature, such as https://doi.org/10.1016/j.ijhydene.2021.09.014, to provide a more comprehensive overview of current state-of-the-art methods.
  5. In Figure 1, Kimtech™ wipes are used for pressure blotting. Given that paper type can significantly influence water removal due to variations in absorbency and fibre structure, could the authors clarify whether different blotting papers were tested or whether the use of Kimtech™ is essential to the reproducibility of the protocol? Is there any validation that the choice of wipe does not introduce variability in the measured water uptake?
  6. Please carefully proofread the manuscript to correct typographical errors and formatting issues. For example, in line 255, "couldconfirm" should be "could confirm," and in the section heading “Cross-validation wither other AEMs,” “wither” should be corrected to “with.”

Author Response

1.Given that AEMs contain reactive functional groups such as quaternary ammonium, has the potential interaction with ambient CO2 been considered? CO2 absorption could lead to the formation of bicarbonate or carbonate species, potentially altering the membrane's ionic form and affecting water uptake. How did the authors account for or minimize this effect during handling and measurement?

Thank you for raising this important point. We fully agree that anion exchange membranes (AEMs) can interact with ambient CO₂, potentially leading to the formation of bicarbonate or carbonate species that could affect their ionic form and, consequently, water uptake.

To minimize this effect, we implemented several measures:

A.Controlled environment: All handling and measurements were conducted in a clean room under controlled conditions—specifically, with constant temperature, humidity, and airflow—to reduce environmental variability and CO₂ exposure.

B. Minimized exposure time: Immediately after soaking, membranes were pressure-blotted and weighed using a balance located directly adjacent to the setup, minimizing the time between hydration and measurement.

C. Consideration of inert atmosphere: We did consider performing the entire procedure in an inert nitrogen atmosphere (e.g., in a glove box) to eliminate CO₂ interaction entirely. However, we ultimately decided against this, as it would have made the protocol more complex and less accessible to reproduce in standard lab settings.

We added the following sentence to our manuscript at the end of the description of the measurement protocol (line 123) In addition, all handling was performed in a clean room with controlled environmental conditions to limit CO₂ exposure, which can affect the ionic form of the membrane

2.When measuring water uptake under different temperature or humidity conditions, the membrane must be tested promptly after being removed from the dish to minimize the influence of ambient water vapor and room temperature on its mass. However, the standardized pressure-blotting method likely takes more time than conventional touch-blotting using Kimtech™ wipes. Has the potential error introduced during this additional handling time been considered by the authors? Could this error be comparable to or even larger than the differences between the two blotting methods themselves? What strategies could be employed to minimize this time-dependent error?

Thank you for this valuable observation. We fully agree that minimizing time between membrane removal and weighing is critical, especially when working under varying temperature or humidity conditions, to avoid unwanted mass changes due to ambient water vapor.

We shared this concern during method development and took care to evaluate the potential impact of handling time. In practice, we found that if all materials (e.g., Kimtech™ wipes and blotting setup) are prepared in advance during the soaking step, the pressure-blotting process adds only a few seconds—resulting in a total handling time of slightly over 20 seconds. This is comparable to the conventional touch-blotting method, especially when considering that repeated swiping may be needed to remove surface water, depending on user technique and applied pressure.

Moreover, in the conventional method, significant variability exists between users—some may complete blotting and weighing in ~10 seconds, while others may take 40 seconds or more. This introduces time as an uncontrolled variable. By contrast, our standardized pressure-blotting method ensures a fixed and reproducible timing sequence, thereby reducing operator-dependent variability.

To minimize time-dependent error, our strategy was to:

A.Prepare all materials and equipment in advance.

B.Position the analytical balance immediately next to the blotting setup to avoid unnecessary delay.

C.Follow a fixed sequence from membrane removal to blotting and weighing, which ensures consistent timing across all measurements.

These steps allowed us to achieve high reproducibility, while controlling for potential time-based mass changes during the procedure.

We added the following sentence in the manuscript after the experimental description (line 123) All materials were prepared in advance, and the membrane was weighed immediately after blotting using a balance positioned directly next to the setup, minimizing time-dependent variation due to ambient humidity or temperature.

3. The standardized pressure-blotting method uses ~0.44 N cm⁻² applied for 10 s. Has the sensitivity of water uptake to blotting pressure been evaluated? Specifically, how robust is the measured uptake to slight variations in applied pressure?

Thank you for your thoughtful question. While we did not perform a systematic sensitivity analysis of water uptake as a function of blotting pressure, we did explore various setups during preliminary tests. These included different pressure levels and application methods. Our aim was to identify conditions that were both practical and yielded reproducible results.

Through this process, we found that a pressure of ~0.44 N cm⁻² applied twice for 10 s provided a good balance—ensuring a uniform pressure distribution, effective removal of excess surface water, and minimal variability between replicates. Once we achieved spot-free membranes and consistent uptake values, we did not further investigate slight variations in pressure, as the method appeared sufficiently robust for our purposes.

4.We thank the reviewer for this thoughtful and constructive suggestion. We reviewed the recommended paper by Zheng et al. (Int. J. Hydrogen Energy, 2021, 46, 37137–37151), which provides valuable insights into the water uptake behavior of ETFE-based AEMs. However, we would like to clarify that this study used a conventional gravimetric immersion method rather than dynamic vapor sorption (DVS).

Nevertheless, we fully agree with the reviewer that DVS represents an advanced and complementary technique for characterizing water sorption and transport under vapor-phase conditions. To acknowledge this and highlight the practical advantages of our method, we have added the following sentence to the Introduction (line 47)

 While advanced methods such as dynamic vapor sorption (DVS) offer detailed insights into vapor-phase hydration behavior, they require specialized equipment, long equilibration times, and are less accessible for routine membrane screening.

We believe this addition provides a balanced perspective on the current state-of-the-art while reinforcing the accessibility and reproducibility of the method presented in our study.

5.In Figure 1, Kimtech™ wipes are used for pressure blotting. Given that paper type can significantly influence water removal due to variations in absorbency and fibre structure, could the authors clarify whether different blotting papers were tested or whether the use of Kimtech™ is essential to the reproducibility of the protocol? Is there any validation that the choice of wipe does not introduce variability in the measured water uptake?

We thank the reviewer for this relevant observation. We did not carry out a systematic investigation of the influence of different blotting papers on water uptake results. Kimtech™ wipes were selected as they are specifically designed for laboratory use, are low-lint and additive-free, and are commonly used in similar membrane studies.

Since our focus was on elaborating and validating a reproducible protocol—including controlled pressure, timing, and handling—we did not extend the scope to testing different paper types. However, we acknowledge that the absorbency characteristics of the blotting material may play a role and agree this could be a valuable subject for future exploration.

6.Please carefully proofread the manuscript to correct typographical errors and formatting issues. For example, in line 255, "couldconfirm" should be "could confirm," and in the section heading “Cross-validation wither other AEMs,” “wither” should be corrected to “with.”

We thank the reviewer for pointing out these typographical and formatting issues. We have carefully proofread the entire manuscript and corrected the noted errors, including changing “couldconfirm” to “could confirm” and “wither” to “with” in the section heading. Additional minor inconsistencies and formatting issues were also corrected throughout the text to improve clarity and readability

Round 2

Reviewer 2 Report

Comments and Suggestions for Authors

The authors have thoroughly addressed all of my concerns. The manuscript is now suitable for publication.